# Focus on the Dynamics of Neutrophil-to-Lymphocyte Ratio in Cancer Patients Treated with Immune Checkpoint Inhibitors: A Meta-Analysis and Systematic Review

**DOI:** 10.3390/cancers14215297

**Published:** 2022-10-27

**Authors:** Yusheng Guo, Dongqiao Xiang, Jiayu Wan, Lian Yang, Chuansheng Zheng

**Affiliations:** 1Department of Radiology, Union Hospital, Tongji Medical College, Huazhong University of Science and Technology, Wuhan 430022, China; 2Hubei Key Laboratory of Molecular Imaging, Wuhan 430022, China

**Keywords:** neutrophil-to-lymphocyte ratio, immune checkpoint inhibitor, meta-analysis, early change

## Abstract

**Simple Summary:**

Neutrophil-to-lymphocyte ratio (NLR) as a prognostic indicator of patients treated with immunotherapy has been fully reported. However, the dynamics of NLR after immunotherapy and its association with efficacy of immunotherapy have been less frequently reported. This meta-analysis and systematic review study aimed to (a) summarize the early change in NLR after immune checkpoint inhibitor (ICI) treatment, (b) evaluate the association between the trend in NLR and efficacy of ICI treatment, and (c) analyze the prognostic value of baseline and post-treatment NLR.

**Abstract:**

Background: A number of studies have reported an association between the dynamics of neutrophil-to-lymphocyte ratio (NLR) and clinical efficacy in patients treated with immune checkpoint inhibitors (ICIs), but there is still a lack of a meta-analysis or systematic review. Methods: PubMed, Embase, Web of Science, and the Cochrane Library were searched until September 2022 for studies reporting on the association between the change in NLR after ICI treatment and clinical outcomes. Outcome measures of interest included: change in NLR before and after treatment, overall survival (OS), progression-free survival (PFS), and objective response rate (ORR). Results: A total of 4154 patients in 38 studies were included. The pooled percentage of patients with increased NLR was 49.7% (95CI%: 43.7–55.8%). Six studies discussing the change in NLR in patients with different tumor responses all showed that the NLR level in patients without response to immunotherapy may increase after ICI treatment. The upward trend in NLR was associated with shorter OS (pooled HR: 2.05, 95%CI: 1.79–2.35, *p* < 0.001) and PFS (pooled HR: 1.89, 95%CI: 1.66–2.14, *p* < 0.001) and higher ORR (pooled OR: 0.27, 95%CI: 0.19–0.39, *p* < 0.001), and downward trend in NLR was associated with longer OS (pooled HR: 0.49, 95%CI: 0.42–0.58, *p* < 0.001) and PFS (pooled HR: 0.55, 95%CI: 0.48–0.63, *p* < 0.001) and lower ORR (pooled OR: 3.26, 95%CI: 1.92–5.53, *p* < 0.001). In addition, post-treatment high NLR was associated with more impaired survival than baseline high NLR (pooled HR of baseline high NLR: 1.82, 95%CI: 1.52–2.18; pooled HR of post-treatment high NLR: 2.93, 95%CI: 2.26–3.81), but the NLR at different time points may have a similar predictive effect on PFS (pooled HR of baseline high NLR: 1.68, 95%CI: 1.44–1.97; pooled HR of post-treatment high NLR: 2.00, 95%CI: 1.54–2.59). Conclusions: The NLR level of tumor patients after ICI treatment is stable overall, but the NLR level in patients without response to immunotherapy may increase after ICI treatment. Patients with an upward trend in NLR after ICI treatment were associated with worse clinical outcomes; meanwhile, the downward trend in NLR was associated with better clinical outcomes. Post-treatment high NLR was associated with more impaired survival than baseline high NLR.

## 1. Introduction

For many decades, surgery, radiotherapy, and chemotherapy were the main therapy for cancer [1]. Recent rapid developments in immunotherapy have attracted increased attention and have transformed practice in the treatment of many types of cancers; of special interest are immune checkpoint inhibitors (ICIs) targeting programmed cell death-1/programmed death ligand-1 (PD-1/PD-L1) and T-lymphocyte-associated protein 4 (CTLA-4) [2]. Despite the progress that has been achieved, the efficacy of ICI treatment is not totally satisfactory because only a part of patients responded to ICI treatment [3], and various immune-related adverse events (irAEs) during the course of therapy were not uncommon [4]. Therefore, it is essential to select appropriate biomarkers to identify the patients who may not benefit from ICI treatment to avoid useless, expensive, and possibly toxic treatments.

However, only PD-L1 expression, tumor mutational burden (TMB), and microsatellite instability (MSI) entered routine clinical practice in patients with non-small-cell lung cancer (NSCLC) [5,6,7]. Next-generation sequencing (NGS) and immunohistochemical analysis were needed to confirm these biomarkers [8]. These tissue-based biomarkers are unsuitable for serial monitoring; meanwhile, biopsy site and specimen status may be bias factors affecting the results. Hence, it is necessary to find more readily available biomarkers suitable for diverse clinical settings (including low-resource settings) to predict the effect of ICI treatment in more types of tumors. The neutrophil-to-lymphocyte ratio (NLR), a blood-based biomarker defined by the absolute counts of neutrophils and lymphocytes, has received more and more attention because of its relatively easy and inexpensive accessibility.

Inflammation is an important characteristic of cancer [9], and neutrophils have various important biological functions in innate and adaptive immunities, thus playing a key role in inflammation [10]. Activated neutrophils also were able to suppress T-lymphocyte function by secreting myeloperoxidase and arginase-1 and upregulating PD-L1, thus resulting in the delivery of a negative signal to T cells [11]. Therefore, neutrophils may help create an immunosuppressive tumor microenvironment which reduces the efficacy of immunotherapy [12]. NLR could serve as a measure of the balance of adaptive immune surveillance and inflammation status; consequently, it was natural to explore the association between NLR and the efficacy of ICI treatment. There have been numerous systematic reviews and meta-analyses demonstrating that a high level of NLR at baseline was associated with poor prognosis in many types of cancers [13,14,15]. However, significant changes in serum cytokine concentrations were observed after the initial ICI treatment, and these changes were likely to be associated with treatment response [16,17].

Therefore, we hypothesized that systemic inflammatory status related to immune responses against tumors may be dynamic rather than static [18], and the early changes in NLR detected from peripheral blood could be the potential biomarkers of tumor burden and predict the very early efficacy of cancer immunotherapy in the initial weeks of ICI treatment before imaging evaluation or when it is not conclusive [16].

In fact, recently published studies about immunotherapy and the dynamics of NLR suggested the special role of early change in NLR in predicting the efficacy of immunotherapy and final survival outcomes [8,18,19,20,21,22,23,24,25,26,27,28,29,30,31,32,33,34,35,36,37,38,39,40,41,42,43,44,45,46,47,48,49,50,51,52,53,54]. Specifically, they observed the following three points: (1) The trend in NLR (upward or downward) may be associated with the outcome of tumor response after ICI treatment. (2) The different trends in NLR appear to be able to stratify the survival times of patients who received immunotherapy. (3) Compared with the baseline NLR, post-treatment high NLR was seemly associated with more impaired survival. Although a considerable number of studies discuss the early change in NLR after immunotherapy and its association with prognosis, there is still a lack of relevant review. Hence, we focused on the novel topic of dynamics of NLR in cancer patients who received ICI treatment, and we conducted this meta-analysis and systematic review study in an effort to comprehensively summarize the change in NLR after ICI treatment and its association with prognosis.

## 2. Methods

### 2.1. Search Strategy

Literature review and data extraction followed established PRISMA guidelines [55], and we prospectively registered the protocol in the PROSPERO International Register of Systematic Reviews (CRD42022313394). We conducted comprehensive literature searches in PubMed, Embase, Web of Science, and the Cochrane Library until 10 September 2022. The search terms were structured by combining the keywords including “neutrophil-to-lymphocyte ratio”, “NLR”, “immune checkpoint inhibitor”, “ICI”, “immunotherapy”, “cancer”, “tumor”, and “solid tumor”. An example of a search strategy used for PubMed is as follows: (NLR or (neutrophil-to-lymphocyte ratio)) and ((immune checkpoint inhibitor) or immunotherapy or ICI) and ((solid tumor) or cancer). After obtaining all results from four databases, we firstly deleted the duplicate results and then completed a manual search of potentially missing studies by screening the references of the studies.

### 2.2. Inclusion and Exclusion Criteria

Two authors (Yusheng Guo and Jiayu Wan) independently conducted the literature search and preliminary screening of the literature from the databases by reading titles and abstracts. Studies meeting the inclusion criteria were included in this study: (1) studies written in English; (2) studies investigating the association between the dynamics of NLR and the prognosis of patients who received ICI treatment; (3) studies reporting overall survival (OS), progression-free survival (PFS), objective response rate (ORR), or the change in NLR before and after treatment. The exclusion criteria were as follows: (1) review studies, meta-analyses, conference or poster abstracts, case reports, comments, letters, and editorials; (2) studies on non-solid tumors; (3) duplicate reports and ongoing studies.

### 2.3. Data Extraction

Two authors independently assessed the quality and the risk of bias in each included study using the Newcastle–Ottawa Scale. Any disagreements were resolved by a third author (Dongqiao Xiang or Lian Yang). The following data were extracted independently by two authors: year of publication, first author, region, treatment, type of tumor, number of patients, the time point for rechecking NLR after immunotherapy, outcome, gender, Newcastle–Ottawa Scale, study design (Table 1). In the case where multiple publications reported overlapping data, the study with the largest sample size was considered.

### 2.4. Statistical Analyses

All statistical analysis was conducted using R software (version 4.1.0). We assessed heterogeneity by using a chi-square test and the I^2^ metric before performing a meta-analysis. The I^2^ value indicates the percentage of variability across the pooled estimates attributable to statistical heterogeneity, and studies with I^2^ > 40% were considered as having high heterogeneity. We used a random effects model if high heterogeneity was present and used a fixed effects model in case of low heterogeneity. Next, forest plots were made to show the HR/OR and 95% CI of each study and the pooled HR/OR and 95% CI. We explored possible sources of heterogeneity by using Baujat plots and sensitivity analyses conducted by excluding the studies one by one. Finally, publication bias was assessed by funnel plot, Egger test, and Begg test. Wherever publication bias was detected, the trim-and-fill method was implemented to produce a model after accounting for any publication bias. Two-sided *p* < 0.05 was considered statistically significant for all statistical procedures.

## 3. Results

We identified 2663 studies in the database searches, and we excluded 2084 duplicated articles. After analysis of the title, abstract, and topic, 541 other articles were excluded (Figure 1). Finally, 38 studies were included (34 studies were included in the meta-analysis for quantitative analyses and 6 studies were included in the systematic review to analyze the dynamics of NLR level in patients without or with response to immunotherapy).

### 3.1. Characteristics of Included Studies

In total, 38 studies involving 4154 patients were included in this meta-analysis and systematic review [8,18,19,20,21,22,23,24,25,26,27,28,29,30,31,32,33,34,35,36,37,38,39,40,41,42,43,44,45,46,47,48,49,50,51,52,53,54]. Thirty-seven studies had a retrospective study design, and only one study was prospective [24]. All the included studies received moderately high scores from the Newcastle–Ottawa Scale quality assessments. Of the 38 studies included, 6 studies [8,37,40,44,48,54] considered two or more types of tumors, and 32 studies dealt with specific types of cancer (non-small-cell lung cancer was the most frequently studied tumor). Seven studies [21,27,30,37,41,44,48] did not mention the specific drugs used, and the most widely used ICI is nivolumab, which was used in 24 studies [8,18,22,23,25,28,29,31,32,33,34,36,38,39,40,42,43,47,50,51,52,53,54]. Time points for rechecking NLR included 2 weeks to 8 weeks, after one to four cycles of treatment, after one to four infusions, and the most common time points were 4 weeks and 6 weeks.

### 3.2. Change in NLR before and after ICI Treatment

First, we investigated the studies that reported the proportion of patients with increased or decreased NLR after ICI treatment. A total of 12 studies [19,20,33,39,41,43,44,47,48,49,51,54] involving 1449 patients reported 726 patients (50.1%) with upward trend in NLR and 723 patients (49.9%) with downward trend in NLR. The proportion of patients with increased NLR ranged from 24.6% to 60.8% in 12 studies, and the pooled percentage of patients with increased NLR was 49.7% (95CI%: 43.7–55.8%) (Appendix A). Although ICI treatment did not influence the trend in NLR overall, six studies [27,32,37,42,45,53] reported that the NLR level in patients without response to immunotherapy tended to significantly increase after ICI treatment, and in contrast, patients who responded to ICI treatment have similar or lower NLR level before and after immunotherapy (Appendix A). Notably, a retrospective study with data from a multicenter prospective trial using a mixed effects regression analysis with per-patient random intercept reported that patients with CR/PR/SD had a lower log-transformed NLR at study entry (*p* = 0.03) and the NLR level was stable during follow-up overall [37], which corresponded to our results.

### 3.3. Different Trends in NLR after ICI Treatment Were Associated with Different Prognoses

Eighteen studies (Sekine et al. reported two independent cohorts) [8,21,22,23,26,29,30,33,34,40,44,45,47,48,50,51,52,54] including 2324 patients looked at the association between OS and the different trends in NLR after ICI treatment. Fifteen studies [8,21,23,26,29,30,33,34,40,44,47,48,50,51,54] reported the relationship between the upward trend in NLR and OS, and the subsequent time points for rechecking NLR included 4–8 weeks, a treatment cycle after ICI treatment, or before the second infusion. Considering the low heterogeneity (I^2^= 9%), a fixed effects model was used for analysis. Results indicated the pooled HR was 2.05 (95%CI: 1.79–2.35, *p* < 0.001), suggesting the negative prognostic role of the upward trend in NLR in patients who received immunotherapy (Figure 2A). Twelve studies (13 cohorts) [21,22,29,30,40,44,45,47,48,51,52,54] investigated the association between the downward trend in NLR and OS. The time points for rechecking included 4–8 weeks, after the first treatment cycle, before the second infusion, or after two doses. Given the low heterogeneity (I^2^ = 27%), a fixed effects model showed that the pooled HR was 0.49 (95%CI: 0.42–0.58, *p* < 0.001, Figure 2B).

Twenty studies (Sekine et al. reported two independent cohorts) [8,18,20,21,22,23,26,28,29,30,33,40,44,45,46,47,48,50,52,54] including 2494 patients reported the association between PFS and the change in NLR after immunotherapy. Fifteen studies [8,18,20,21,23,26,29,30,33,40,44,46,47,50,54] reported the association between the increase in NLR after ICI treatment and PFS. The pooled HR from a fixed effects model was 1.89 (95%CI: 1.66–2.14, *p* < 0.001, I^2^ = 37%, Figure 2C). Fourteen studies (15 cohorts) [20,21,22,28,29,30,33,40,44,45,47,48,52,54] reported the association between the increase in NLR after ICI treatment and PFS. There was low heterogeneity (I^2^ = 2%) in these studies, and a fixed effects model indicated that the decrease in NLR was associated with longer PFS (pooled HR: 0.57, 95%CI: 0.50–0.65, *p* < 0.001, Figure 2D).

In addition, we investigated the association between the trends in NLR after ICI treatment and ORR, and 10 studies [19,20,21,28,30,39,41,43,45,47] provided data for ORR. Data from eight studies [19,20,21,30,39,41,43,47] on the increase in NLR after ICI treatment and ORR indicated that the upward trend in NLR after immunotherapy was significantly associated with worse tumor response (pooled OR: 0.27, 95%CI: 0.19–0.39, *p* < 0.001, I^2^ = 8%, Figure 2E). Meanwhile, a random effects model pooling OR from 10 studies [19,20,21,28,30,39,41,43,45,47] showed that the downward trend in NLR after immunotherapy was associated with better tumor response (pooled OR: 3.26, 95%CI: 1.92–5.53, *p* < 0.001, I^2^ = 56%, Figure 2F). The Baujat plot showed that the study by Wang et al. contributed the maximum heterogeneity and influence to the overall result (Appendix A). The results of the sensitivity analysis did not change after excluding the studies one by one (Appendix A). Appendix A in the Appendix A lists more details about these studies.

### 3.4. Post-Treatment High NLR Was Associated with More Impaired Survival Than Baseline High NLR

Among the previous studies, a considerable number of studies have demonstrated the negative impact of the elevation of the baseline (pre-treatment) NLR on the prognosis. Next, we compared the impact of NLR at baseline and after treatment on prognosis, based on those studies that gave both time points at the same time (Appendix A).

Twelve studies [24,25,29,30,31,35,36,38,39,40,45,53] reported the association between OS and NLR at two different time points with the same cut-off values of NLR. Given the low heterogeneity (I^2^ = 19%), a fixed effects model was used for analysis, and the results showed that a high level of NLR at baseline was associated with poor overall survival (pooled HR: 1.82, 95%CI: 1.52–2.18, *p* < 0.001, Figure 3A). Meanwhile, a random effects model indicated that post-treatment high NLR level was associated with poor survival outcomes (pooled HR: 2.93, 95%CI: 2.26–3.81, *p* < 0.001, I^2^ = 45%, Figure 3A). The Baujat plot indicated that the biggest source was from the study by Wang et al. (Appendix A), and the sensitivity analysis found similar results (Appendix A). It was worth noting that post-treatment high NLR was associated with more impaired survival than baseline high NLR.

Eleven studies [24,29,30,31,35,36,38,40,43,45,53] investigated the association between PFS and NLR at two different time points with the same cut-off values of NLR. A fixed effects model indicated that baseline high NLR level was associated with shorter PFS (pooled HR: 1.68, 95%CI: 1.44–1.97, *p* < 0.001, I^2^ = 0%, Figure 3B). Moreover, a random effects model showed that post-treatment high NLR level was associated with shorter PFS (pooled HR: 2.00, 95%CI: 1.54–2.59, *p* < 0.001, I^2^ = 48%, Figure 3B). Sensitivity analysis demonstrated the stability of these results (Appendix A). However, the NLR at different time points may have a similar predictive effect on PFS.

### 3.5. Publication Bias

Potential publication bias was assessed using funnel plots, the Egger test, and the Begg test. All funnel plots were approximately symmetrical (Figure 4A–J), and the results of the Egger test demonstrated that there was a publication bias in the studies reporting the association between the upward trend in NLR and PFS (Table 2). The trim-and-fill method showed that it was necessary to fill four potential unpublished studies in the funnel plot (Appendix A); the result was not relevantly changed with a pooled HR of 1.86 (95%CI: 1.58–2.19, *p* < 0.001).

## 4. Discussion

People have been looking for inexpensive and reproducible biomarkers that can accurately predict responses to immune checkpoint inhibition. Compared with biomarkers such as PD-L1, TMB, and MSI, routine blood samples are more readily available and do not require any other additional costs; thus, they have been easily applied in the real-world setting. [56]. More importantly, the prognostic role of NLR seems to be applicable to most patients with various tumors receiving ICI treatment [57,58,59]; therefore, it is necessary to conduct a further meta-analysis on NLR.

All previous meta-analyses about immunotherapy and NLR focused on the impact of baseline NLR on prognosis [60,61,62,63,64,65]. Although a small part of the meta-analyses included a few studies addressing the dynamics of NLR, they did not conduct quantitative analysis and did not arrive at a specific conclusion [60,61]. At present, this study is the first meta-analysis and systematic review focusing on the dynamics of NLR in cancer patients after ICI treatment and the relationship between the dynamics of NLR and prognosis. Our study yielded three key findings. First, we observed that ICI treatment was not associated with a significantly increased or decreased NLR level in the overall cohort; however, it seems that among patients who did not respond to immunotherapy, NLR may have a significant upward trend. Second, different trends in NLR after ICI treatment were associated with different prognoses. Results indicated that the upward trend in NLR was associated with worse clinical outcomes and that the downward trend in NLR was associated with better clinical outcomes. Finally, post-treatment high NLR associated was with more impaired survival than baseline high NLR.

Whether in daily clinical practice or clinical trials, imaging is often the standard to judge whether the tumor progression exists or not. However, patients often have their first imaging evaluation 6–12 weeks after treatment [66,67,68,69]. Imaging evaluation could be considered earlier when the NLR is relatively increased after ICI treatment, especially in patients with high levels of baseline or post-treatment NLR. Moreover, recently published studies indicated NLR (including baseline NLR and early change in NLR) is an important factor for hyperprogressive disease and pseudo-progression, suggesting the combination of NLR and imaging evaluation may help in reaching more accurate conclusions when imaging evaluation alone is not conclusive [33,70,71].

It is recognized that many components, including stromal cells, immune cells, and vasculature, together constitute a complex tumor microenvironment [72]. It has been widely reported that in the immune cells, tumor-infiltrating lymphocytes play an important role in cancer immune surveillance and cytotoxic cell death and therefore inhibit the growth of tumors [73]. Checkpoint inhibitor therapies cannot come into play in patients with immunologically cold tumors which contain scarce tumor-infiltrating lymphocytes [74]. Previous studies indicated tumor cells secrete chemokines and growth factors such as IL-8 and granulocyte-colony stimulating factor recruiting neutrophils into tumors, aiding vascular invasion, and orchestrating the metastatic potential of tumor cells [42,75]. More importantly, neutrophils contribute to forming the immunosuppressive microenvironment by secreting myeloperoxidase and arginase-1 and upregulating PD-L1 and MDSC, a differentiation status of suppressive myeloid cells, preventing T-cell activation in the tumor and resulting in a decrease in the efficacy of immunotherapy [11]. The correlation between peripheral blood NLR and clinical outcomes may be attributed to the correlation between circulating neutrophils and neutrophils in the tumor microenvironment [76,77], and low levels of circulating lymphocytes may be associated with low levels of tumor-infiltrating lymphocytes and reduced anti-tumor T-cell responses [12,78,79]. A high level of NLR at baseline treatment implies an increase in neutrophil count and/or a decrease in lymphocyte count, probably indicating the repertoires of anti-tumor immunity are relatively absent; meanwhile, the occurrence of increased NLR after ICI treatment is more likely to imply the absence of response to ICI treatment, finally impairing survival benefit of patients.

This analysis has some limitations. First, most of the included studies are retrospective studies leading to inevitable selection bias. Second, there is no unified definition of critical value for the increase or decrease in NLR; this is also a potential source of heterogeneity. In addition, our study included a limited category of malignancies, mainly including NSCLC, renal cell carcinoma, and melanoma. Therefore, further large-scale prospective studies including more types of malignancies are needed. Despite these limitations, the results of the meta-analysis are reliable because among most of the results, low heterogeneity was detected and publication bias was not observed.

## 5. Conclusions

The NLR level of tumor patients after ICI treatment is stable overall, but the NLR level in patients without response to immunotherapy may increase after ICI treatment. Patients with an upward trend in NLR after ICI treatment were associated with worse clinical outcomes; meanwhile, the downward trend in NLR was associated with better clinical outcomes. Post-treatment high NLR was associated with more impaired survival than baseline high NLR. Monitoring the dynamics of NLR in patients treated with immunotherapy may contribute to the evaluation of tumor response, risk stratification, and patient management.

## Figures and Tables

**Figure 1 cancers-14-05297-f001:**
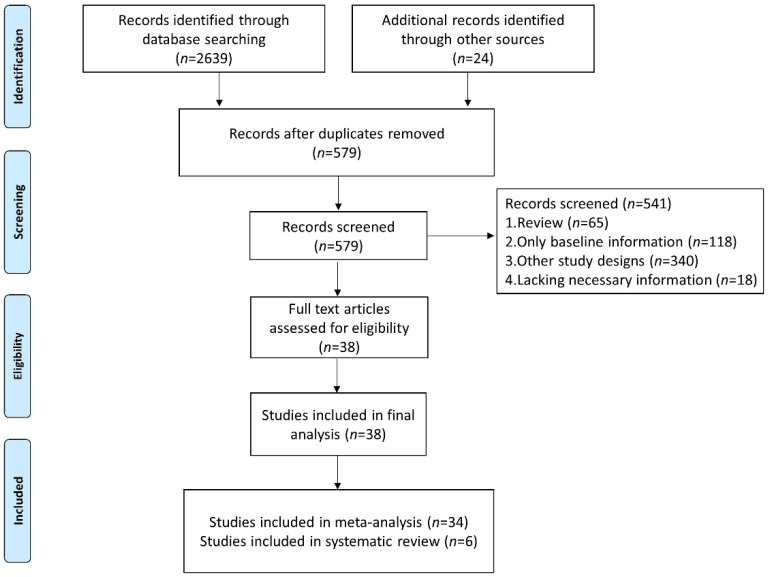
Flow diagram of study selection for inclusion in this meta-analysis and systematic review.

**Figure 2 cancers-14-05297-f002:**
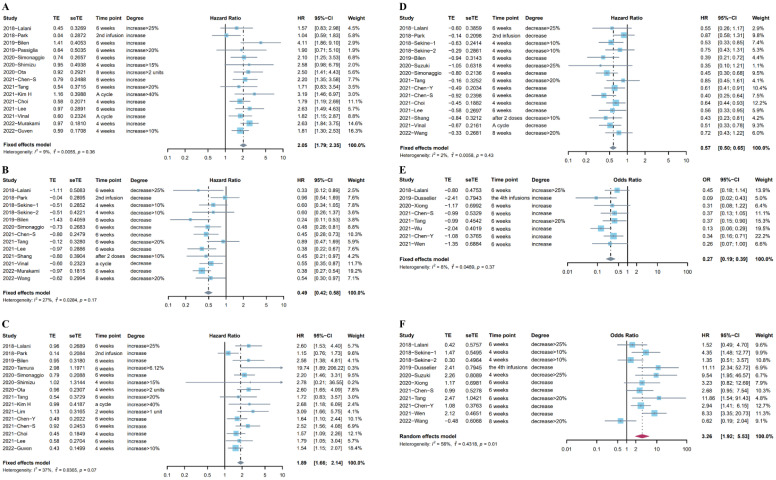
(**A**) Forest plot for the association between upward trend in NLR after ICI treatment and OS. (**B**) Forest plot for the association between downward trend in NLR after ICI treatment and OS. (**C**) Forest plot for the association between upward trend in NLR after ICI treatment and PFS. (**D**) Forest plot for the association between downward trend in NLR after ICI treatment and PFS. (**E**) Forest plot for the association between upward trend in NLR after ICI treatment and ORR. (**F**) Forest plot for the association between downward trend in NLR after ICI treatment and ORR.

**Figure 3 cancers-14-05297-f003:**
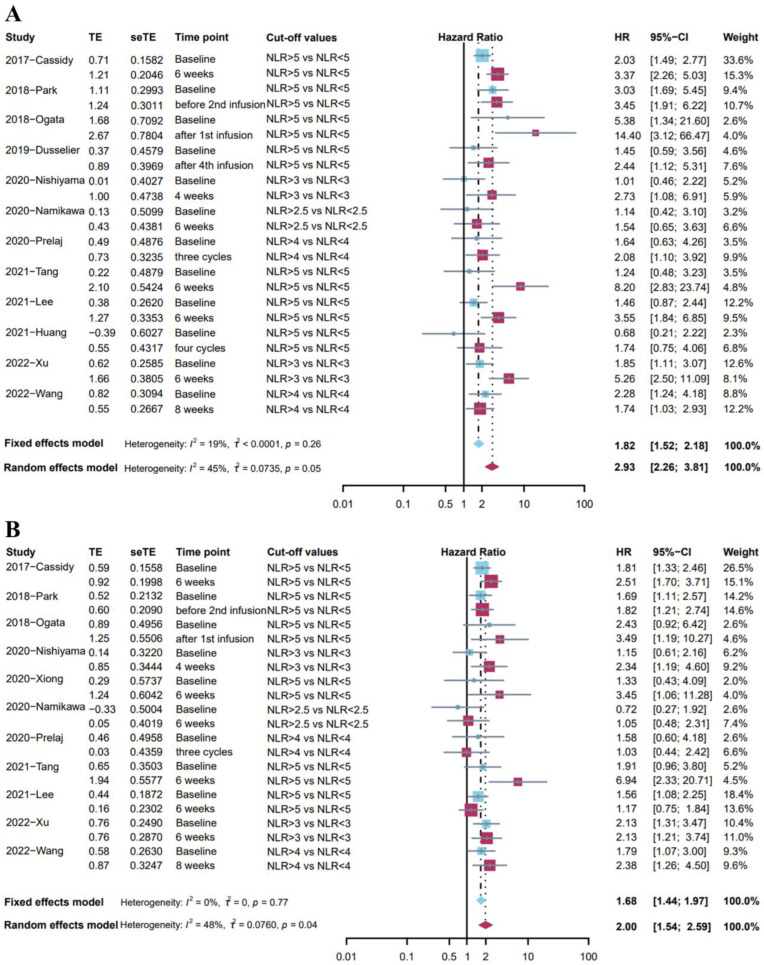
(**A**) Forest plot for the association between high level of baseline NLR and OS (blue); the association between high level of post-treatment NLR and OS (red). (**B**) Forest plot for the association between high level of baseline NLR and PFS (blue); the association between high level of post-treatment NLR and PFS (red).

**Figure 4 cancers-14-05297-f004:**
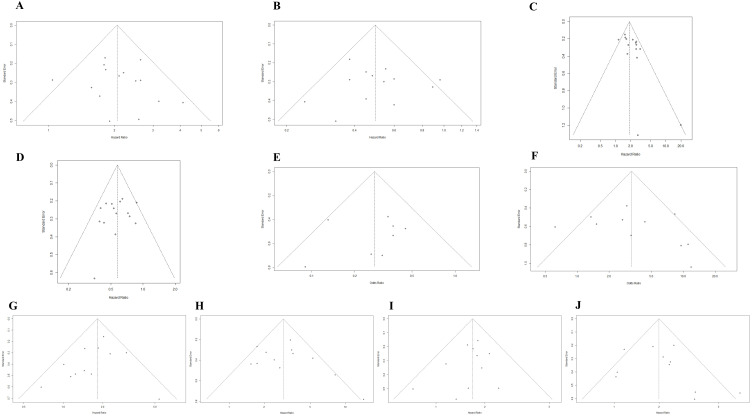
(**A**) Funnel plot for the association between upward trend in NLR after ICI treatment and OS. (**B**) Funnel plot for the association between downward trend in NLR after ICI treatment and OS. (**C**) Funnel plot for the association between upward trend in NLR after ICI treatment and PFS. (**D**) Funnel plot for the association between downward trend in NLR after ICI treatment and PFS. (**E**) Funnel plot for the association between upward trend in NLR after ICI treatment and ORR. (**F**) Funnel plot for the association between downward trend in NLR after ICI treatment and ORR. (**G**) Funnel plot for the association between high level of baseline NLR and OS. (**H**) Funnel plot for the association between high level of post-treatment NLR and OS. (**I**) Funnel plot for the association between high level of baseline NLR and PFS. (**J**) Funnel plot for the association between high level of post-treatment NLR and PFS.

**Table 1 cancers-14-05297-t001:** Characteristics of included studies.

Year	Author	Region	Treatment	Tumor	Number of Patients	Time Point After Immunotherapy	Outcome	Gender (Male/Female)	Newcastle–Ottawa Scale	Study Design	Reference Number
2017	Michael R Cassidy	USA	Ipilimumab	Melanoma	197	3 weeks and 6 weeks	OS; PFS	125/72	9	Retrospective	[35]
2018	Aly-Khan A Lalani	USA	Anti-PD-1 and anti-PD-L1	mRCC	192	6 weeks	OS; PFS; ORR	101/41	9	Retrospective	[21]
2018	Katsutoshi Sekine	Japan	Nivolumab (Cohort 1)	NSCLC	87	4 weeks	OS; PFS; ORR	54/33	8	Retrospective	[22]
2018	Katsutoshi Sekine	Japan	Nivolumab (Cohort 2)	NSCLC	75	4 weeks	OS; PFS; ORR	50/25	8	Retrospective	[22]
2018	Takatsugu Ogata	Japan	Nivolumab	Gastric cancer	26	Two weeks after the first administration	OS; PFS; ORR	19/7	6	Retrospective	[38]
2018	Alona Zer	Canada	Anti-PD-L1	NSCLC	88	8 weeks	Change in NLR before and after treatment	43/45	7	Retrospective	[27]
2018	Malaka Ameratunga	The United Kingdom, Spain, and Australia	Anti-PD-1 and anti-PD-L1	Advanced solid tumors	165	Each cycle of therapy	Change in NLR before and after treatment	91/74	9	Retrospective	[37]
2018	Wungki Park	USA	Nivolumab	NSCLC	159	Before the second nivolumab infusion	OS; PFS	82/77	7	Retrospective	[29]
2018	Monica Khunger	USA	Nivolumab	NSCLC	109	After 2 cycles of treatment	Change in NLR before and after treatment	56/53	7	Retrospective	[42]
2019	Mehmet A Bilen	USA	/	Melanoma, gastrointestinal cancer, lung/head and neck cancer, breast cancer, and others	90	6 weeks	Change in NLR before and after treatment; OS; PFS	53/37	8	Retrospective	[44]
2019	Francesco Passiglia	Italy	Nivolumab	NSCLC	45	6 weeks	OS	32/13	7	Retrospective	[34]
2019	Matthieu Dusselier	France	Nivolumab	NSCLC	59	The 4th nivolumab infusions	Change in NLR before and after treatment	44/15	7	Retrospective	[39]
2020	Arsela Prelaj	Italy	Nivolumab and pembrolizumab	NSCLC	154	The second cycle and third cycle	OS; PFS	126/28	7	Retrospective	[31]
2020	Kotaro Suzuki	Japan	Nivolumab	mRCC	65	4 weeks	OS; PFS; ORR	47/18	7	Retrospective	[28]
2020	A Simonaggio	France	Nivolumab	mNSCLC and mRCC	161	6 weeks	Change in NLR before and after treatment; OS; PFS	114/47	9	Retrospective	[54]
2020	Takuto Shimizu	Japan	Pembrolizumab	Metastatic urothelial carcinoma	27	4 weeks	OS; PFS	23/4	6	Retrospective	[26]
2020	Yumiko Ota	Japan	Nivolumab	Gastric cancer	98	4 weeks (PFS) and 8 weeks (OS)	OS; PFS	68/30	7	Retrospective	[23]
2020	Naotaka Nishiyama	Japan	Nivolumab	mRCC	52	4 weeks	OS; PFS	36/16	7	Retrospective	[36]
2020	Lan Huang	China	Nivolumab, pembrolizumab, atezolizumab and ipilimumab	NSCLC	61	The fourth cycle of treatment	OS; PFS	38/23	7	Retrospective	[25]
2020	Tsutomu Namikawa	Japan	Nivolumab	Gastric cancer	29	2 weeks, 4 weeks, 6 weeks, and 8 weeks	OS; PFS	19/10	6	Retrospective	[53]
2020	Daichi Tamura	Japan	Pembrolizumab	Urothelial carcinoma	41	6 weeks	PFS	29/12	6	Retrospective	[46]
2020	Daiki Ikarashi	Japan	Nivolumab	mRCC	45	6 weeks	Change in NLR before and after treatment	30/15	6	Retrospective	[32]
2021	Yin Tang	China	Anti-PD-1 and anti-PD-L1	NSCLC	124	6 weeks	OS; PFS; ORR	89/35	8	Retrospective	[30]
2021	Xianbin Wu	China	Anti-PD-1	Esophageal squamous cell carcinoma	119	6 weeks	Change in NLR before and after treatment; ORR	102/17	7	Retrospective	[41]
2021	Jeong Uk Lim	Korea	Nivolumab, pembrolizumab, and atezolizumab	NSCLC	89	2 weeks	PFS	62/27	7	Retrospective	[18]
2021	Yuzhong Chen	China	Pembrolizumab, Sintilimab and Toripalimab	NSCLC	151	6 weeks and 12 weeks	Change in NLR before and after treatment; OS; PFS; ORR	115/36	8	Retrospective	[20]
2021	Won-Mook Choi	Korea	Nivolumab	HCC	194	2 weeks, 4 weeks, and 6 weeks	Change in NLR before and after treatment; OS; PFS	159/35	8	Retrospective	[33]
2021	Shixue Chen	China	Nivolumab, pembrolizumab, and others	NSCLC	101	6 weeks	Change in NLR before and after treatment; OS; PFS; ORR	72/29	8	Retrospective	[47]
2021	Pei Yi Lee	Singapore	Nivolumab, pebrolizumab, atezolimumab, avelumab, durvalumab, and tremelimumab	Lung cancer, colorectal cancer, nasopharyngeal carcinoma, gastric cancer, hepatocellular carcinoma	147	6 weeks	OS; PFS	99/48	7	Retrospective	[40]
2021	Jin Shang	China	Nivolumab, pembrolizumab, atezolimab, ipilimumab, and sintilimab	Pancreatic Cancer	122	After 2 doses	OS; PFS	87/35	7	Retrospective	[52]
2021	Qi Xiong	China	Nivolumab, pembrolizumab, atezolizumab, and toripalimab	SCLC	41	6 weeks	Change in NLR before and after treatment; OS; PFS; ORR	36/5	7	Retrospective	[43]
2021	D Viñal	Spain	/	Lung cancer, melanoma, kidney cancer, bladder cancer, others	211	A treatment cycle	Change in NLR before and after treatment; OS; PFS	136/75	7	Retrospective	[48]
2021	Yoshiaki Yamamoto	Japan	Pembrolizumab	Urothelial carcinoma	121	6 weeks	Change in NLR before and after treatment	87/34	7	Retrospective	[49]
2021	Jwa Hoon Kim	Korea	Nivolumab and pembrolizumab	Esophageal squamous cell carcinoma	60	A treatment cycle	OS; PFS	56/4	7	Retrospective	[50]
2022	Shaodi Wen	China	Pembrolizumab, sintilimab, toripalimab	NSCLC	90	6 weeks and 12 weeks	Change in NLR before and after treatment; ORR	69/21	7	Retrospective	[19]
2022	Yusuke Murakami	Japan	Nivolumab	NSCLC	162	4 weeks	Change in NLR before and after treatment; OS	111/51	8	Retrospective	[51]
2022	Lin Wang	China	Camrelizumab	Esophageal squamous cell carcinoma	69	8 weeks	OS; PFS; ORR	64/5	8	Retrospective	[45]
2022	Jianming Xu	China	Sintilimab	Esophageal squamous cell carcinoma	97	6 weeks	OS; PFS	88/7	8	Prospective	[24]
2022	Deniz Can Guven	Turkey	Nivolumab, atezolizumab, pembrolizumab, ipilimumab, and avelumab	RCC, melanoma, NSCLC and others	231	4 weeks	OS; PFS	155/76	7	Retrospective	[8]

Note. USA: United States of America; NSCLC: non-small-cell lung cancer; SCLC: small cell lung cancer; mNSCLC: metastatic non-small-cell lung cancer; RCC: renal cell carcinoma; mRCC: metastatic renal cell carcinoma; HCC: hepatocellular carcinoma; NLR: neutrophil-to-lymphocyte ratio; OS: overall survival; PFS: progression-free survival; ORR: objective response rate; PD-1: programmed cell death-1; PD-L1: programmed death ligand-1.

**Table 2 cancers-14-05297-t002:** Publication bias.

Description	*p*-Value of Egger Test	*p*-Value of Begg Test	Corresponding Funnel Plot	Corresponding Forest Plot
The association between the upward trend in NLR and OS	0.5542	0.2550	Figure 4A	Figure 2A
The association between the downward trend in NLR and OS	0.8197	1.0000	Figure 4B	Figure 2B
The association between the upward trend in NLR and PFS	0.0184	0.0536	Figure 4C	Figure 2C
The association between the downward trend in NLR and PFS	0.4286	0.7290	Figure 4D	Figure 2D
The association between the upward trend in NLR and ORR	0.6975	0.3223	Figure 4E	Figure 2E
The association between the downward trend in NLR and ORR	0.4106	0.4835	Figure 4F	Figure 2F
The association between the baseline NLR and OS	0.2993	0.1702	Figure 4G	Figure 3A
The association between the post-treatment NLR and OS	0.3730	0.3370	Figure 4H	Figure 3A
The association between the baseline NLR and PFS	0.3288	0.3115	Figure 4I	Figure 3B
The association between the post-treatment NLR and PFS	0.4003	0.3115	Figure 4J	Figure 3B

Note. NLR: neutrophil-to-lymphocyte ratio; OS: overall survival; PFS: progression-free survival; ORR: objective response rate.

## Data Availability

The data used and/or analyzed for this study are available from the corresponding author upon reasonable request.

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
