# Peer review of "Focus on the Dynamics of Neutrophil-to-Lymphocyte Ratio in Cancer Patients Treated with Immune Checkpoint Inhibitors: A Meta-Analysis and Systematic Review"

_cancers, 2022, doi:10.3390/cancers14215297_

Round 1

Reviewer 1 Report

In the present systemic review authors aim to evaluate  the dynamics of NLR in cancer patients after ICI treatment and the relationship between the dynamics of NLR and prognosis. They conducted detailed PRISMA compliant systematic review and meta-analysis. This is a challenging topic for which there is no systematic review. The author team summarized the most recent and relevent papers on the subject. Although there are several typos and the narrative is a kind of fragmentary, I just abide by scientific soundness. The text is well written and very easy to read and follow it. I would like to offer the following points for consideration by the authors towards the improvement of the manuscript:

1-  Authors must give more information about the dynamics of NLR in cancer patients in the introduction section.

2- Authors must provide more detailed search strategy, especially keywords. Even when searching only "neutrophil-to-lymphocyte ratio" AND "cancer" in Pubmed database, there are 2810 studies

3- Authors mentioned  that review studies, meta-analyses, conference or poster summaries, case reports, comments, letters, and editorials as exclusion criteria. However, it is specified only as review in the flow diagram.

4- Please clearly indicate the exact systematic search dates in Methods.

5- I think it would be better to add baseline and dynamic NLR parameters to the table.

Author Response

In the present systemic review authors aim to evaluate the dynamics of NLR in cancer patients after ICI treatment and the relationship between the dynamics of NLR and prognosis. They conducted detailed PRISMA compliant systematic review and meta-analysis. This is a challenging topic for which there is no systematic review. The author team summarized the most recent and relevent papers on the subject. Although there are several typos and the narrative is a kind of fragmentary, I just abide by scientific soundness. The text is well written and very easy to read and follow it.

R1-1: Authors must give more information about the dynamics of NLR in cancer patients in the introduction section.

Response: We highly appreciate the reviewer’s constructive comments. We have added more give more information about the dynamics of NLR in cancer patients in the introduction section. We described the three points found by previous studies to explain why we should conduct this work: 1) The trend of NLR (upward or downward) may be associated with the outcome of tumor response after ICI treatment. 2) The different trends of NLR could stratify the survival times of patients who received immunotherapy. 3) Compared with the value of baseline NLR, post-treatment high NLR was seemly associated with more impaired survival. These three points may help improve the readability to readers (have a quick impression on previous studies) and interest in reading this paper. In addition, we checked and corrected typos in this manuscript.

R1-2: Authors must provide more detailed search strategy, especially keywords. Even when searching only "neutrophil-to-lymphocyte ratio" AND "cancer" in PubMed database, there are 2810 studies

Response: We highly appreciate the reviewer’s comment. We performed literature search through combining more keywords in four databases. For example, we performed the search with the “(NLR or (neutrophil-to-lymphocyte ratio)) and ((immune checkpoint inhibitor) or immunotherapy) and ((solid tumor) or cancer)”and got only 598 results (2022.10.19) instead of 2810 results with combining "neutrophil-to-lymphocyte ratio" AND "cancer". Again, thank this reviewer for his/her constructive comment. We have provided a more detailed description of search strategy in the section “Search strategy”, as requested.

R1-3: Authors mentioned that review studies, meta-analyses, conference or poster summaries, case reports, comments, letters, and editorials as exclusion criteria. However, it is specified only as review in the flow diagram.

Response: We appreciate the reviewer’s advice. In fact, “Other study designs” in the flow diagram included the “meta-analyses, conference or poster summaries, case reports, comments, letters, and editorials”. Again, thank this reviewer for his/her comment.

R1-4: Please clearly indicate the exact systematic search dates in Methods.

Response: Yes, the exact systematic search date has been added in the subsection “Search strategy” of the “Methods” section, as requested.

R1-5: I think it would be better to add baseline and dynamic NLR parameters to the table.

Response: We highly appreciate the reviewer’s constructive comments. In a single table, Table 1 was too dense to fill other 2-3 parameters. Therefore, we have added Table S2-3 which summarized the baseline NLR and dynamic NLR parameters. 

Reviewer 2 Report

The paper of Guo et al. filled a gap in the literature regarding the use of an easy accessed biomarker in patients on ICI treatment.

The authors stated clearly what study found and how they did it. The research question also justified given what is already known about the topic.

The study methods are valid and reliable. Statistically significant results are clear. Results are discussed from different angles and placed into context without being overinterpreted.

The conclusions answer the aim of the study. The conclusions are supported by references and own results.

The limitations of the study are not fatal, but they are opportunities to inform future research.

 Specific comments on weaknesses of the article and what could be improved:

Major points  

1.          The biological significance and possible clinical application of NLR as a blood-based biomarker for cancer should be explained with at least a paragraph in the introduction.

2.          The paper needs stronger conclusions and recommendations for the clinical practice

Minor points

1.           Please, don't start sentences with a number (i.e., in the Abstract - " 6 studies discussing the change of NLR in patients..."

2.           Could you consider inserting the reference number for each study in Table 1?

Author Response

The paper of Guo et al. filled a gap in the literature regarding the use of an easy accessed biomarker in patients on ICI treatment. The authors stated clearly what study found and how they did it. The research question also justified given what is already known about the topic. The study methods are valid and reliable. Statistically significant results are clear. Results are discussed from different angles and placed into context without being overinterpreted. The conclusions answer the aim of the study. The conclusions are supported by references and own results. The limitations of the study are not fatal, but they are opportunities to inform future research.

R2-1: (Major point 1) The biological significance and possible clinical application of NLR as a blood-based biomarker for cancer should be explained with at least a paragraph in the introduction.

Response: We thank the reviewer for this constructive comment. We supplemented content in paragraph 2, 3, 4, and 5 of the section “Introduction” about the biological significance and possible clinical application of NLR, as requested.

R2-2: (Major point 2) The paper needs stronger conclusions and recommendations for the clinical practice

Response: Yes, we have supplemented content in the conclusion. Here we highlight that monitoring the dynamics of NLR in patients treated with immunotherapy may contribute to the evaluation of tumor response, risk stratification, and patient management. According to irRECIST [1], confirming hyperprogressive disease or pseudo-progression always relied on the continues observation if the first imaging evaluation was immune unconfirmed progressive disease (iUPD). Therefore, the best time to change the therapy may be missed.  So, we supplemented content in the discussion emphasizing that the combination of NLR and imaging evaluation may help come to more accurate conclusions when only imaging evaluation was not conclusive.

Reference:

1.Seymour L, Bogaerts J, Perrone A, Ford R, Schwartz LH, Mandrekar S, et al. iRECIST: guidelines for response criteria for use in trials testing immunotherapeutics. The Lancet Oncology. 2017;18(3):e143-e52.

R2-3:(Minor point 1) Please, don't start sentences with a number (i.e., in the Abstract - " 6 studies discussing the change of NLR in patients..."

Response: We highly appreciate the reviewer’s comment. We have corrected it (starting sentences with a number) as requested.

R2-4: (Minor point 2) Could you consider inserting the reference number for each study in Table 1?

Response: We highly appreciate the reviewer’s constructive comments. We listed the reference number for each study in Table 1. In addition, we also listed the reference number for studies in Table S1-3 for the convenience of the reader.

Round 2

Reviewer 1 Report

I am satisfied that the authors have addressed all of my previous concerns about the article. It is now much improved and I feel that it is now suitable for publication.